# Adjusting Organic Load as a Strategy to Direct Single-Stage Food Waste Fermentation from Anaerobic Digestion to Chain Elongation

**Vicky De Groof** [1,4] **, Marta Coma** [3] **, Tom C. Arnot** [2,3,4] **, David J. Leak** [2,3,5] **and Ana B. Lanham** [2,3,4,*]

1    EPSRC Centre for Doctoral Training in Sustainable Chemical Technologies, University of Bath, Claverton Down, Bath BA2 7AY, UK; V.De.Groof@bath.ac.uk
2    Water Innovation & Research Centre (WIRC), University of Bath, Claverton Down, Bath BA2 7AY, UK; T.C.Arnot@bath.ac.uk (T.C.A.); D.J.Leak@bath.ac.uk (D.J.L.)
3    Centre for Sustainable and Circular Technologies (CSCT), University of Bath, Claverton Down, Bath BA2 7AY, UK; M.Coma@bath.ac.uk
4    Department of Chemical Engineering, University of Bath, Claverton Down, Bath BA2 7AY, UK
5    Department of Biology & Biochemistry, University of Bath, Claverton Down, Bath BA2 7AY, UK
*    Correspondence: A.Lanham@bath.ac.uk; Tel.: +44-122-538-4544

**Abstract:** Production of medium chain carboxylic acids (MCCA) as renewable feedstock bio-chemicals, from food waste (FW), requires complicated reactor configurations and supplementation of chemicals to achieve product selectivity. This study evaluated the manipulation of organic loading rate in an un-supplemented, single stage stirred tank reactor to steer an anaerobic digestion (AD) microbiome towards acidogenic fermentation (AF), and thence to chain elongation. Increasing substrate availability by switching to a FW feedstock with a higher COD stimulated chain elongation. The MCCA species n-caproic (10.1 ± 1.7 g L$^{-1}$) and n-caprylic (2.9 ± 0.8 g L$^{-1}$) acid were produced at concentrations comparable to more complex reactor set-ups. As a result, of the adjusted operating strategy, a more specialised microbiome developed containing several MCCA-producing bacteria, lactic acid-producing *Olsenella* spp. and hydrogenotrophic methanogens. By contrast, in an AD reactor that was operated in parallel to produce biogas, the retention times had to be doubled when fed with the high-COD FW to maintain biogas production. The AD microbiome comprised a diverse mixture of hydrolytic and acidogenic bacteria, and acetoclastic methanogens. The results suggest that manipulation of organic loading rate and food-to-microorganism ratio may be used as an operating strategy to direct an AD microbiome towards AF, and to stimulate chain elongation in FW fermentation, using a simple, un-supplemented stirred tank set-up. This outcome provides the opportunity to repurpose existing AD assets operating on food waste for biogas production, to produce potentially higher value MCCA products, via simple manipulation of the feeding strategy.

**Keywords:** acidogenic fermentation; anaerobic digestion; food waste; medium chain carboxylic acids; microbial chain elongation; mixed culture; organic loading rate

## 1. Introduction

To achieve a sustainable and circular bio-economy, it is crucial to minimise food waste (FW, 88 M tonnes in EU annually) and use the unavoidable, inedible fraction as feedstock for the production of bio-chemicals [1,2]. FW is rich in carbon, nutrients, and moisture, making it a favourable substrate for mixed microbial culture fermentation, such as anaerobic digestion (AD) [3]. Recently, research has focused on the carboxylate platform where the liquid intermediates formed during the primary

acidogenic fermentation (AF) steps of AD are targeted to generate products with a higher value than biogas [4]. Among the different compounds that can be obtained, medium-chain carboxylic acids (MCCA) are of particular interest due to their lower water solubility, which facilitates their recovery, their antimicrobial properties, and their potential application as platform chemical or liquid drop-in biofuels [5–7].

Selective operational conditions in AF allow to direct the product outcome of FW towards, e.g., volatile fatty acids (VFA) [8,9], lactic acid [10,11] or hydrogen [12,13]. Some bacteria in AF can elongate short VFA into MCCA with 6 (n-caproic acid) to 8 (n-caprylic acid) carbon atoms via the reversed β-oxidation pathway [14]. Selectivity towards chain elongation is subject to the absence of competitive pathways and the availability of electron donors such as hydrogen, lactic acid, or ethanol [15]. External addition of electron donors is generally not desirable as it has associated costs and a negative impact on the environmental life cycle assessment of waste fermentation for MCCA production [16]. Finding the most suitable operational parameters to direct the complex network of biochemical reactions in mixed culture fermentation towards chain elongation is still a topic of research.

One of the factors currently limiting MCCA yields is competition with methane generation, i.e., transformation of soluble organics into gaseous products. Strategies proposed to block methanogenesis, without the addition of chemical inhibitors, include inoculum pre-treatment by heat shock to select for spore-forming bacteria [17,18], which are mainly fermentative, or lowering the sludge retention time (SRT) to wash out methanogens [8]. Additionally, methanogens are generally more sensitive to a lower pH (<6.0) and the accumulation of VFA that follows from an organically overloaded bioreactor [19]. Organic overload can be obtained in batch operation by increasing the loading rate or food-to-microorganism ratio (F/M > 1 $gCOD_{fed}$ $gVS_{inoculum}^{-1}$) [20].

At low pH and high product concentrations, the carboxylate product species shift towards their acidic, undissociated forms. These acidic compounds have antimicrobial properties and can slow down or inhibit metabolism [21]; hence, chain elongation has been improved by alleviating product toxicity via in situ extraction of MCCA [22–24]. Elongation is also improved by operating at low hydraulic retention time (HRT) and thus low product concentrations, while maintaining production rates, in systems with biomass retention, such as in an up-flow anaerobic sludge blanket reactor [25], or granular sludge reactors [26,27]. Some of these solutions require high recirculation flow rates, which can prove challenging for substrates with high solid concentrations (e.g., >6% *w/w* total solids), such as FW. Alternatively, MCCA production from these types of feedstock has been improved by using two-stage systems. Hydrolysis and acidogenesis can be optimised in a separate bioreactor to chain elongation [28–30], or by using leach-bed reactors where soluble, inhibitory monomers are removed from the solid substrate [31]. Recent work has found that hydrothermal and ultrasonic pre-treatment of FW can enhance MCCA production in fermentation [32]. Such adaptions are a trade-off between achieving higher yields and the costs related to more complicated operation and reactor design. For these reasons, AD reactor configurations at a commercial scale in waste valorisation facilities are typically single-step systems, such as single-stage stirred-tank reactors (STR), as they allow simpler processing and lower investment costs [33,34]. It is, therefore, valuable to explore the potential of such simpler reactor setups for MCCA production from FW to facilitate commercial implementation.

Lab-scale trials have demonstrated that, for maize and switch grass stillage as feedstock, long-term MCCA formation can be achieved in STRs, without the need for addition of electron donors [35,36]. This was due to the in-situ production of lactic acid as electron donor during the fermentation process. A similar mechanism was found for short-term sequential batch fermentation of FW [37]. However, the long-term conversion of FW to MCCA in a STR setup without electron donor supplementation has yet to be demonstrated. In addition, these studies inoculated their reactors with enriched microbiomes and hence the strategy required to operationally transform functionality from a microbiome performing AD to one performing chain elongation remains largely unknown. Therefore, the aim was to demonstrate how an AD microbiome fermenting FW in a simple STR can be redirected, during long-term operation, towards either biogas or MCCA production, by solely

changing operating conditions, predominantly organic overload, i.e., without addition of methanogenic inhibitors or electron donors. This simple approach allows repurposing of existing food waste AD assets to produce higher value products (MCCA), which is an attractive means for accelerating the deployment of circular economy practices. To allow direct comparison of functionality and microbial community development, two STRs, one for AD and the other for AF, were operated in parallel with the same inoculum and fed with the same FW substrate.

## 2. Materials and Methods

Food waste (FW) and inoculum were sourced from a full-scale industrial AD plant (GENeco, Bristol, UK). The FW in the AD plant comprises packaged and unpackaged Category 3 FW collected from households, supermarkets, restaurants and other catering services, and is ground and mixed with a variety of liquid streams from the food-processing industry and/or the liquid fraction of anaerobic digester effluent, to form a slurry-like mixture. Two batches of FW slurry were collected one month apart (FW 1 and FW 2). Upon collection each batch of FW was characterised (Table 1) and frozen in aliquots ($-18\,^{\circ}C$) for reactor feeding. The inoculum was collected from the effluent of a mesophilic continuous anaerobic digester ($2400\ m^3$, STR) processing pasteurised FW. The inoculum was diluted with tap water to reach a set VS in the reactors and acclimated overnight to operating temperature before initiating the feed.

**Table 1.** Characteristics of the two food waste substrates (FW 1 and FW 2) used in this study. Analysis was performed in duplicate and presented with standard deviation.

| Parameter | FW 1 | FW 2 |
|---|---|---|
| pH | 5.0 ± 0.1 | 5.4 ± 0.1 |
| Conductivity (mS cm$^{-1}$) | 6.20 | 6.16 |
| Solid Content (% *w/w*) | | |
| Total Solids | 9.94 ± 0.07 | 17.7 ± 1.6 |
| Volatile Solids | 8.832 ± 0.002 | 16.3 ± 1.1 |
| Chemical Oxygen Demand (gCOD L$^{-1}$) | | |
| Total COD | 150 ± 1 | 297 ± 9 |
| Soluble COD | 37.4 ± 1.1 | 38.2 ± 0.1 |
| Soluble Compounds (g L$^{-1}$) | | |
| Acetic acid | 1.27 ± 0.18 | 0.89 ± 0.16 |
| n-Propionic acid | 0.63 ± 0.02 | 0.63 ± 0.19 |
| n-Butyric acid | <0.31 | 0.35 ± 0.24 |
| MCCA (C5–C8) | 0.00 ± 0.00 | 0.00 ± 0.00 |
| Glucose | 3.61 ± 0.14 | 5.03 ± 0.87 |
| Sugar compounds * | 0.55 ± 0.07 | 5.34 ± 2.13 |
| Lactic acid | 7.27 ± 0.34 | 2.48 ± 0.02 |
| Ethanol | 1.39 ± 0.71 | 0.80 ± 0.16 |

* fructose, overlap with sucrose and xylose; % *w/w*: mass fraction; MCCA (C5–C8): medium chain carboxylic acids (chain length of 5 to 8 carbon atoms).

Two 2 L STRs were operated semi-continuously (1 L working volume, magnetic stirrer mixing, $35\,^{\circ}C$). Feeding events took place every 3.5 days, where a fixed volume of reactor effluent, determined by the set OLR, was manually replaced by the same amount of FW. One STR was set up for AD by starting operation with a F/M of 0.8 gCOD gVS$^{-1}$ and 20 gVS L$^{-1}$ of inoculum, feeding at an average OLR of 4.2 ± 0.4 gCOD L$^{-1}$ d$^{-1}$ (2.5 ± 0.2 gVS L$^{-1}$ d$^{-1}$). The second STR was set up for AF by organic overload at a F/M ratio of 8.4 gCOD gVS$^{-1}$ and 5 gVS L$^{-1}$ of inoculum, feeding at an average OLR of 8.5 gCOD L$^{-1}$ d$^{-1}$ (5.0 gVS L$^{-1}$ d$^{-1}$). Following start-up, reactors were operated in two distinct phases according to the conditions in Table 2. The pH was manually corrected to a minimum of 7.3 ± 0.1 for the AD reactor, or 6.0 ± 0.2 for the AF reactor with sodium hydroxide (1 or 2 M) after each substrate addition. Reactors were operated as airtight, but at intervals were briefly open to atmosphere during feeding and pH correction.

<div align="center">

**Table 2.** Overview of operational parameters for the AD and AF reactors.

</div>

| STR | Feedstock | Days | OLR (gCOD L$^{-1}$ d$^{-1}$) | HRT (d) |
|---|---|---|---|---|
| | Phase 1—Shift functionality with increased organic load | | | |
| AD | FW 1 | 0–14 | 4.2 ± 0.4 | 35 ± 3 |
| | FW 2 | 14–32 | 8.5 ± 0.8 | 35 ± 3 |
| AF | FW 1 | 0–14 | 8.5 ± 0.7 | 18 ± 2 |
| | FW 2 | 14–32 | 17.1 ± 1.5 | 18 ± 2 |
| | Phase 2—Establish longer term operation | | | |
| AD | FW 2 | 0–80 | 4.4 ± 0.5 | 69 ± 6 |
| AF | FW 2 | 0–10 | NA * | NA * |
| | FW 2 | 10–87 | 21.3± 1.6 | 14 ± 1 |

* Gradual start-up from Phase 1 AF after reactor pause where OLR increased from 9.2 to 21.3 gCOD L$^{-1}$ d$^{-1}$ (HRT decrease from 32 to 14 days).

The F/M was determined at start-up and each point of feeding as the amount of total COD (tCOD) fed, over the volatile solids (VS) concentration in the reactors at that time (Equation (1)).

$$F/M_{(i)} = (C_{feed(i)} \times V_{feed(i)})/VS_{reactor(i)} \text{ [gCOD gVS}^{-1}\text{]} \tag{1}$$

where i represents the feeding event, $C_{feed}$ is the organic content expressed as tCOD in the feed in gCOD L$^{-1}$ and V represents volume in L. The OLR was calculated as an average between feeding events as a proxy for continuous operation as the amount of total chemical oxygen demand (tCOD) fed, over the time in between feedings in days per reactor volume (Equation (2)).

$$OLR_{(i)} = (C_{feed(i)} \times V_{feed(i)})/(V_{reactor(i)} \times (t_{(i+1)} - t_{(i)})) = C_{feed(i)}/HRT_{(i)} \text{ [gCOD L}^{-1} \text{ d}^{-1}\text{]} \tag{2}$$

where t represents the day of reactor operation, and $t_{(i+1)} - t_{(i)}$ is the time between feeding points.

AD performance was determined from the methane yield, i.e., volume of methane produced at Standard Temperature and Pressure (STP, 273.15 K and 100 kPa) in between feeding events over the amount of substrate fed, expressed in VS or tCOD. The performance of AF was assessed by: (i) the average net production rate (NP$_{CA}$), i.e., the increase of a given carboxylic acid (CA) in the effluent expressed as COD and corrected for feedstock content (Equation (3)); (ii) the average net yield (Y$_{CA}$), i.e., NP$_{CA}$ over OLR (Equation (4)); and (iii) selectivity (S$_{CA}$) of carboxylic acid formation, i.e., the NP$_{CA}$ of a specific single CA over the net production rate of all carboxylic acids expressed as COD (Equation (5)).

$$NP_{CA(i)} = (C_{CA(i), \text{ effluent}} - C_{CA(i-1), \text{ feed}})/HRT_{(i-1)} \text{ [gCOD L}^{-1} \text{ d}^{-1}\text{]} \tag{3}$$

$$Y_{CA(i)} = NP_{CA(i)}/OLR_{(i-1)} \text{ [\%]} \tag{4}$$

$$S_{CA(i)} = NP_{CA(i)}/\sum NP_{CA(i)} \text{ [\%]} \tag{5}$$

Waste stabilization was evaluated by the removal efficiency of VS or tCOD over the respective load (Equation (6)).

$$VS_{rem} = (VS_{feed} - VS_{effluent})/VS_{feed}, \text{ tCOD}_{rem} = (tCOD_{feed} - tCOD_{effluent})/tCOD_{feed} \text{ [\%]} \tag{6}$$

Total solids (TS) and VS were determined according to Standard Methods 2540 G [38]. The COD was assessed with cuvette tests (LCK014, LCI400, Hach, Dusseldorf, Germany) before and after filtration (0.45 µm) for tCOD and soluble COD (sCOD), respectively.

Liquid samples were taken from the reactor effluent before each feeding event to follow process performance. Carboxylic acids (chain length 2 to 8) were measured by a method adapted from Manni and Caron (1995) using gas chromatography (GC, 7890B, Agilent Technologies, Santa Clara, CA,

USA) equipped with a DB-FFAP 122-3232 column (30 m × 0.25 mm × 0.25 m; Agilent Technologies) with a flame ionization detector (FID) [39]. Liquid samples were conditioned with sulphuric acid, sodium chloride and 2-methyl hexanoic acid as internal standard for quantification before extraction with diethyl ether. Samples (1 μL) were injected at 250 °C with a split ratio of 10, 3 mL min$^{-1}$ purge flow, and $N_2$ carrier gas at 2.4 mL min$^{-1}$ flow rate. The oven temperature was increased by 8 °C min$^{-1}$ from 110 °C to 165 °C, where it was kept for 2 min and the FID temperature was set at 300 °C. The FW samples were further characterised for ethanol, lactic acid, and sugars by high pressure liquid chromatography (HPLC, 1260 Infinity, Agilent Technologies, Santa Clara, CA, USA) as in Coma et al. [16] with the oven temperature adjusted to 65 °C.

Volumetric biogas production was evaluated by determining the displacement of acidified water (pH < 4.3, HCl) in calibrated glass columns connected to the reactor headspace and reported at STP. Biogas samples were collected from the glass columns just before effluent withdrawal and feed addition. $CH_4$ and $CO_2$ were measured by GC (7890A, Agilent Technologies, Santa Clara, CA, USA) with a HP-PLOT/Q column (Agilent Technologies, Santa Clara, CA, USA), whereby $CH_4$ was detected by FID and $CO_2$ was detected with a thermal conductivity detector (TCD) [34]. $H_2$, $N_2$ and $O_2$ percentages were determined by GC-TCD (3800GC, Varian, Agilent Technologies, Santa Clara, CA, USA) equipped with a molecular sieve column (13 × 60–80 mesh, 1.5 m × 1/8″ × 2.0 mm) with a run time of 1 min. Injection, column, and TCD were set at 250, 40, and 200 °C, respectively. Argon was used as the carrier gas at total flow rate of 75 mL min$^{-1}$. Calibration was carried out with multiple injections of a mixture containing permanent gases at 1%. Gas composition was corrected for air intrusion assuming biogas produced comprised only $CH_4$, $CO_2$ and $N_2$ and normalised to 100%.

Biomass samples for community analysis were taken in duplicate on Day 77 from AD and Day 84 from AF reactor (Phase 2). Samples were stored at −18 °C and processed by DNAsense (Aalborg, North Jutland, Denmark). In short, DNA was extracted using the FastDNA Spin kit for Soils (MP Biomedicals, Solon, OH, USA) [40]. The 16S rRNA gene region V4 sequencing libraries were prepared by an Illumina-based custom protocol [41]. PCR amplifications were done with 515FB (5′-GTGYCAGCMGCCGCGGTAA-3′) and 806RB (5′-GGACTACNVGGGTWTCTAAT-3′) as primer pair to cover both archaeal and bacterial domains [42]. The amplicons were paired-end sequenced (2 × 300 bp) on a MiSeq sequencer (Illumina, San Diego, CA, USA). Forward and reverse reads were prepared for use in the UPARSE workflow [43–45]. The reads were clustered into operational taxonomic units (OTUs) at 97% similarity. Taxonomy was assigned using the RDP classifier in QIIME (80% confidence cut-off) and the SILVA database (release 132) [46–48]. The results were analysed in R (v. 3.5.1, https://www.r-project.org/, 2018) through the Rstudio IDE using the ampvis package (v.2.5.8) and the DNAsense app (DNAsense, Aalborg, North Jutland, Denmark) [40,49]. Sequences have been deposited with the ENA database (accession number PRJEB39281). Rarefaction curves, relative abundances, alpha-diversity measures, and taxonomic classifications for all samples are made available within a comprehensive dataset on the University of Bath Research Data Archive [50].

## 3. Results

### 3.1. Elevated Organic Load Directed Anaerobic Digester Sludge towards Acidogenic Fermentation

A start-up strategy of higher F/M and OLR, and hence indirectly lower HRT compared to traditional AD, led in the AF reactor to a net production of carboxylic acids with minimal biogas generation. After two weeks (less than one HRT), VFA (acetic (C2), n-propionic (C3) and n-butyric acid (C4)), accumulated to a total 19.0 gCOD L$^{-1}$ and n-valeric (C5) and n-caproic (C6) acid to 6.1 gCOD L$^{-1}$. The biogas had an average composition of 77 ± 1% $CO_2$, 23 ± 1% $H_2$ and <1% $CH_4$, (Figure 1). In contrast, the operational strategy in the AD reactor resulted in conventional anaerobic digestion of the FW with a methane yield of 0.32 m$^3$ $CH_4$ kgVS$^{-1}$ (0.19 m$^3$ $CH_4$ kgCOD$^{-1}$) on Day 14. This falls within the range of values reviewed for the anaerobic digestion of FW [3].

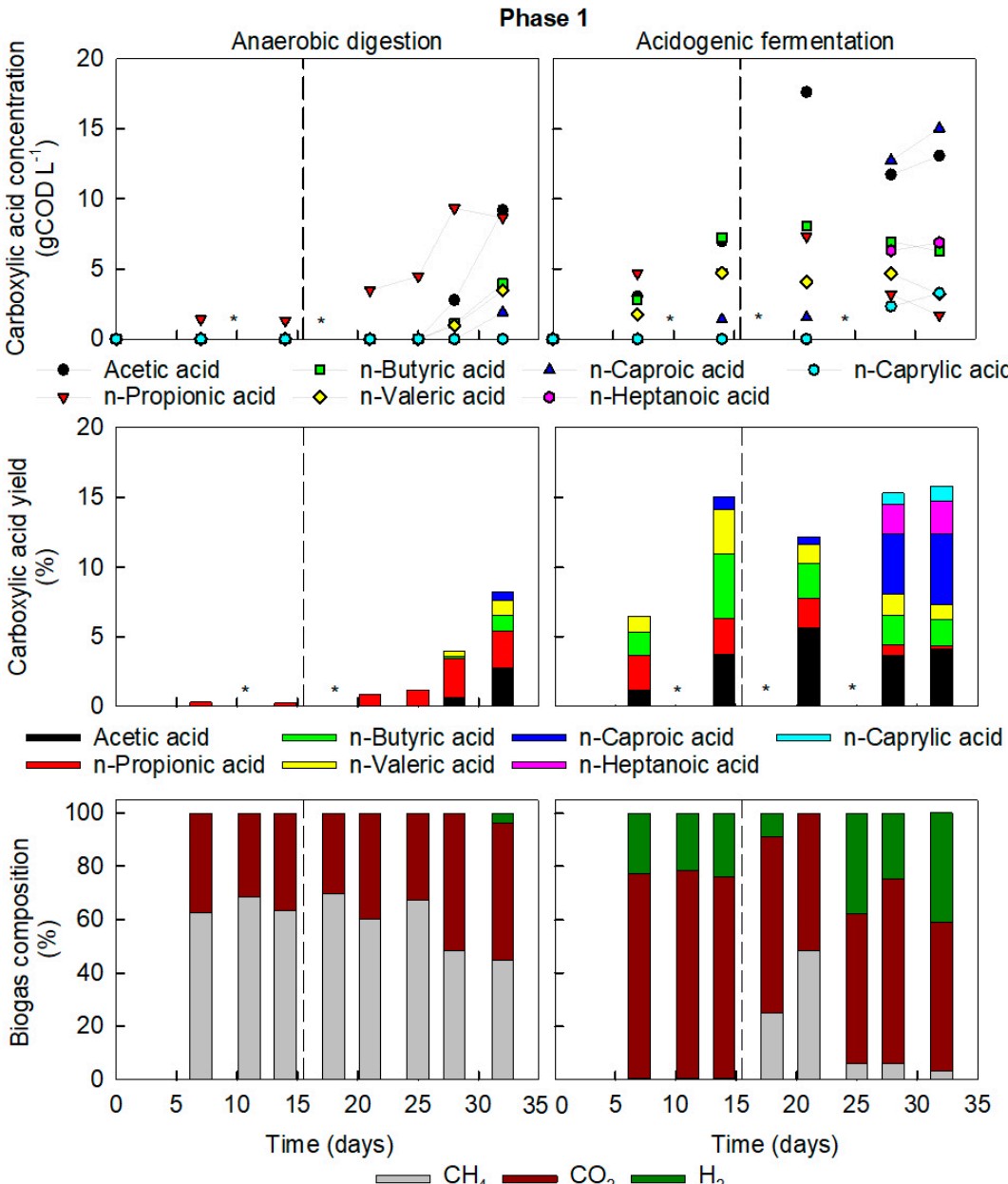

**Figure 1.** Key chemical compounds in Phase 1 of reactor operation in AD (**left**) and AF (**right**). Concentrations (**top**) and yields (**middle**) of liquid fermentation compounds and biogas composition (**bottom**). Dashed lines represent change of feed from FW1 to FW2. * Not determined.

Due to the increased COD content of FW2, the OLR doubled from Day 14 for each reactor while maintaining the HRT. The OLR increased to $17 \pm 2$ gCOD $L^{-1}$ $d^{-1}$, in the AF reactor and to $8.5 \pm 0.8$ gCOD $L^{-1}$ $d^{-1}$ in the AD reactor. In the AF reactor production of carboxylic acids increased and chain elongation occurred up to caprylic acid (C8). By Day 32 MCCA (C6, heptanoic (C7) and C8) totalled 25.2 gCOD $L^{-1}$, i.e., 54% selectivity of all carboxylic acids (Figure 1). Both feedstocks, FW1 and FW2, contained electron donors for chain elongation, but with FW1 containing slightly more ethanol and nearly three times more lactic acid than FW2. Therefore, with the shift from FW1 to FW2, the loading rate of electron donors decreased from $0.60 \pm 0.09$ gCOD $L^{-1}$ $d^{-1}$ to $0.25 \pm 0.02$ gCOD $L^{-1}$ $d^{-1}$, nevertheless chain elongation was stimulated.

The doubling of OLR in AD increased F/M to 1.7 gCOD $gVS^{-1}$, i.e., higher than typical AD values (1 gCOD $gVS^{-1}$) [20]. After two weeks of operating at the elevated OLR, which was similar

to the initial OLR of AF at start-up, the pH dropped to 6.0 in between feeding events. This caused methanogenesis to decrease, with less biogas production and a reduction of methane to 45% of biogas composition, with a resulting yield of 0.02 m$^3$ CH$_4$ kgVS$^{-1}$. Carboxylic acids accumulated simultaneously, reaching concentrations similar to those found during the start-up of AF reactor, namely 21.9 gCOD L$^{-1}$ VFA and 5.4 gCOD L$^{-1}$ of C5 and C6 (Figure 1).

Total biogas production in AF remained at least 9 times lower than the AD reactor. Despite the high organic load, some methanogenic activity did persist in AF. From Day 7 to 14, less than 1% of CH$_4$ was detected in the biogas, but it peaked on Day 21 to 48.4% CH$_4$ (Figure 1). During the previous period, an increase in organic solids concentration in the reactor was observed, from 5 to 22 gVS L$^{-1}$, lowering F/M to 1.38 gCOD gVS$^{-1}$ on Day 14, with a lower F/M enhancing methanogenesis. This could be due to the accumulation of substrate particles or biomass growth. By Day 32 methanogenesis subsided again to 3.1% CH$_4$ content in the off gas probably due to a consistent overload from FW2 with higher COD.

To verify whether methanogensis could recover, no fresh substrate was added to either reactor for 2 weeks (equivalent to 4 feeding events) and pH was corrected. The pH dropped again from 7.1 to 5.5 regardless of the absence of fresh organic material. The overall carboxylic acid content did not reduce, and methanogenic activity did not recover. The high carboxylates concentrations in AD before starvation (27.3 gCOD L$^{-1}$, C2–C6) continued to inhibit methanogenesis as they were far above inhibitory levels, even at neutral pH, i.e., approximately 9.5 gCOD L$^{-1}$ [51,52]. Similarly, in the AF reactor, methane in the biogas remained low (≈1% CH$_4$).

### 3.2. High-COD Food Waste Required Increased Retention Times for AD but Promoted Chain Elongation in AF

The AD reactor was restarted like Phase 1 with fresh inoculum and operated in parallel as control during Phase 2 of operation. The HRT increased to 69 ± 7 days to compensate for the increased tCOD content of the feedstock. The AD reactor was giving methane yields of 0.51 m$^3$ CH$_4$ kgVS$^{-1}$ by Day 24, which was similar to Phase 1. Thus, the AD operation could be adapted for the COD-rich substrate by moving to a longer HRT. After 56 days of operation, i.e., less than one HRT, VFA increased again peaking at 7.3 gCOD L$^{-1}$ and methane yield reduced to 0.17 m$^3$ CH$_4$ kgVS$_{added}^{-1}$ (Figure 2). The increase in HRT could therefore only temporarily restore the AD functionality when applying the high COD-FW. Microbial community analysis revealed that methanogens were still present and thus, were likely inhibited by the VFA. The OLR of 4.4 ± 0.5 gCOD L$^{-1}$ d$^{-1}$ (2.4 ± 0.3 gVS L$^{-1}$ d$^{-1}$) is near the upper limit for stable mono-digestion of FW (2.5 gVS L$^{-1}$ d$^{-1}$) [3].

In Phase 2, operation of AF was resumed and maintained over five HRT to evaluate the long-term effects of an elevated OLR on product outcome and community enrichment. The average OLR in AF was gradually increased after starvation over 4 feeding events from 9.2 to 21 ± 2 gCOD L$^{-1}$ d$^{-1}$, slightly higher than at the end of Phase 1. The increase in OLR resulted in an accumulation of carboxylic acids averaging 48 ± 7 gCOD L$^{-1}$, similar to the end of Phase 1, but with a larger fraction of C5-C8 (73 ± 8%). On resuming operation, C2 immediately decreased, followed by a decline in C4 four feeding events later (Figure 2). The drop in short VFA was accompanied by an increase in C6 and C8, indicating chain elongation. The simple STR setup used in the current study, analogous to current industrial AD setups, resulted in C6 and C8 concentrations of 10.1 ± 1.7 g L$^{-1}$ (22.3 ± 3.6 gCOD L$^{-1}$) and 2.9 ± 0.8 g L$^{-1}$ (7.2 ± 2.0 gCOD L$^{-1}$), respectively, averaged over five HRT. The maximum concentration of undissociated C6 (with antimicrobial properties) was 2.3 g L$^{-1}$ (Day 17 - Phase 2, pH 5.55, total C6 of 13.8 g L$^{-1}$). This is far above the reported inhibitory concentration of 0.87 g L$^{-1}$ [53].

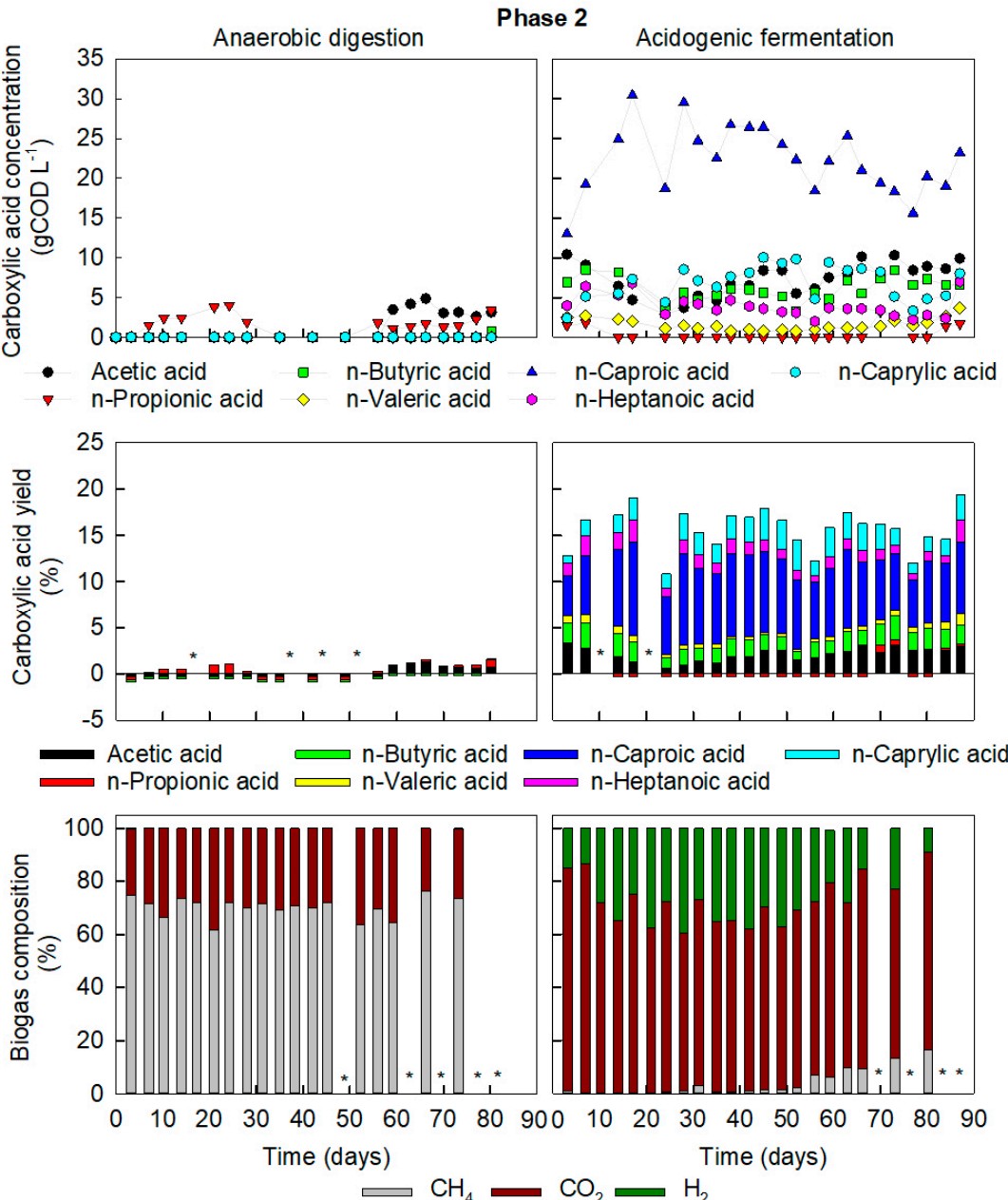

**Figure 2.** Key chemical compounds in Phase 2 of reactor operation in AD (**left**) and AF (**right**). Concentrations (**top**) and yields (**middle**) of liquid fermentation compounds and biogas composition (**bottom**). * Not determined.

By increasing OLR in Phase 2, the loading rate of electron donors, i.e., ethanol and lactic acid present in FW2, slightly rose from $0.25 \pm 0.02$ gCOD $L^{-1}$ $d^{-1}$ to $0.31 \pm 0.03$ gCOD $L^{-1}$ $d^{-1}$. However, the net production rate of MCCA nearly doubled from $1.3 \pm 0.1$ gCOD $L^{-1}$ $d^{-1}$ to $2.4 \pm 0.5$ gCOD $L^{-1}$ $d^{-1}$, so the improved chain elongation could not have been due to electron donors in the influent alone, further indicating stimulation of their in situ production by increased OLR.

All carboxylates with an uneven carbon chain length decreased, with C3 dropping below detection levels, followed by a decline in C5 and C7 concentrations (Figure 2). C3 production from lactic acid through the acrylate pathway is characterised as a competitive pathway of chain elongation and occurs at high concentrations of lactic acid and at a pH above 6 [54–56]. It is hypothesised that by operating at a pH between 5.5 and 6.0 in AF, C3 production was minimized due to increased chain elongation

consuming lactic acid. The limited presence of C3 resulted in a higher selectivity for MCCA with even numbers of carbon (C6 and C8).

At the start of Phase 2, the F/M in AF operation dropped below 1 gCOD gVS$^{-1}$, likely due to solid accumulation, yet $CH_4$ in the biogas remained below 2% (with the rest being $CO_2$ and $H_2$). From Day 56 onwards, the methane fraction increased again and reached a maximum of 17% on Day 80, despite carboxylic acids and OLR being far above accepted values for inhibition of methanogenesis.

### 3.3. Presence of Hydrogen and pH Stabilisation Indicate Chain Elongation

To gain an insight into the cascade of reactions occurring in AD and AF in between feeding events, gas production and carboxylic acid concentrations were followed by regular sampling between two feeding events (Phase 2, Days 21 to 24) (Figure 3). For the AD reactor, the pH profile steadily increased, as often observed with methane production. Only C3 was present in AD, which remained relatively constant (less than 7% change from time 0). Methane production showed a batch-like production profile with a maximum production rate obtained within the first 24 hours from readily biodegradable matter, followed by a slower production rate as further bioavailable matter was consumed.

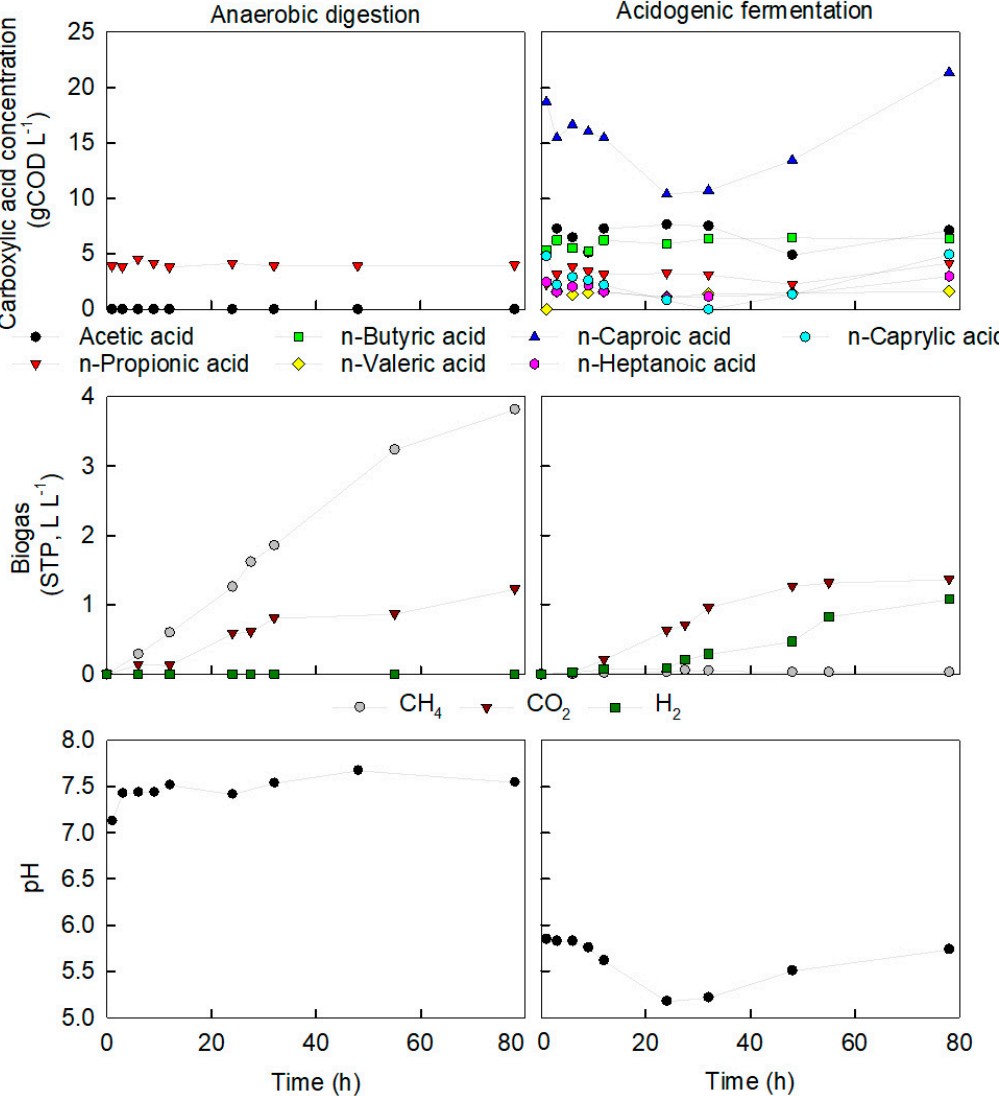

**Figure 3.** Concentration of carboxylic acids (**top**), biogas production (**middle**) and pH profile (**bottom**) in between two feeding events (Day 21 and 24 of Phase 2) for AD (**left**) and AF (**right**). Time 0 corresponds to a sample taken straight after feed addition.

In the AF reactor, the pH decreased from 5.8 to a minimum of 5.2 during the first 32 h after feeding. This is in line with primary acidogenic fermentation where short VFA accumulate and acidification occurs. C5–C8 concentrations were nearly halved while C2–C4 increased by 32%. The reactor headspace had to be opened to introduce feed, thus reducing the $pH_2$ near 0 atm. An elevated $H_2$ partial pressure ($pH_2$), higher than 0.003 atm, must be maintained to ensure a sufficiently reductive environment and avoid the degradation of MCCA into short VFA via the β-oxidation pathway [53]. In between feeding events, the reactors were kept airtight and after 32 hours of primary fermentation, the $pH_2$ in AF headspace reached approximately 0.21 atm. The $H_2$ could have been produced by various metabolic pathways such as primary fermentation, C4 fermentation, acetogenic activity where C2 is converted to $CO_2$ and $H_2$ and within the first step in chain elongation, namely ethanol and lactic acid oxidation [4,14,57,58]. In the following 46 h, pH increased again to 5.7, and $H_2$ and MCCA increased, indicating a secondary fermentation stage of chain elongation [37]. These consecutive fermentation stages where an initial acidification stage is followed by chain elongation is similar to that reported for other chain elongation studies [37,57]. It can be hypothesised that this metabolism was the same in our reactor, although we were unable to analyse the lactic acid and ethanol concentrations to confirm it in this case.

### 3.4. A Distinct Enriched Microbiome for Chain Elongation

The effect of reactor operation on the enrichment of the microbial community was evaluated at the end of Phase 2 (Figure 4). An average of 159 unique observed operational taxonomic units (OTUs) was found in the AD reactor, whereas only an average of 104 OTUs was observed in AF. Similarly, alpha-diversity, richness and evenness measures were lower for the AF microbiome than for AD (Table 3). The AD reactor showed a high relative abundance of Firmicutes (43 ± 2%), Bacteroidetes (18.8 ± 0.1%), Euryarchaeota (10.3 ± 0.1%), and other phyla that are commonly found in anaerobic digesters processing FW (Figure 4) [59,60]. Several hydrolytic and acidogenic groups were detected, such as the proteolytic Firmicutes *Gallicola* (10 ± 2%) and *Fastidiosipila* (12 ± 1%), and the lactic acid-producing *Enterococcus* (1.3 ± 0.3%). The detected bacteria belonging to Bacteroidetes and Synergistetes are generally important hydrolysers in AD that degrade carbohydrates and proteinaceous substrates and contribute to acidogenesis by producing VFA and other acids [61–64]. This is in line with stabilization of waste streams by reducing the solid and organic content of the feedstock in AD, with an average VS and COD removal of 84 ± 9% and 83 ± 5%, respectively, similar to other FW AD studies (42 to 95%) [65]. The various genera that play crucial roles in hydrolysis and acidogenic fermentation present in AD could not be found in in the AF reactor, although some other genera were detected. In the AF reactor, the VS removal only achieved 36 ± 21% in accordance with a lower hydrolytic community. Meanwhile, COD removal in AF accounted for 28 ± 16%, as most of the organics were retained as VFA instead of degassed via methane, although the relative abundance of the archaeal community was similar to that in AD, as discussed below.

**Table 3.** Alpha-diversity indices for the microbial community in the AD and AF reactors. Averaged over duplicate samples and calculated based on 20,314 reads per sample.

| Index | AD | AF |
|---|---|---|
| Observed OTUs | 159 ± 4 | 102 ± 7 |
| Shannon | 3.49 ± 0.04 | 1.70 ± 0.08 |
| Simpson | 0.945 ± 0.007 | 0.66 ± 0.04 |
| InvSimpson | 18.6 ± 0.8 | 2.9 ± 0.3 |
| Chao1 (richness) | 182 ± 16 | 129 ± 5 |
| ACE (richness) | 184 ± 18 | 135 ± 14 |
| Pielou evenness | 0.625 ± 0.007 | 0.31 ± 0.01 |

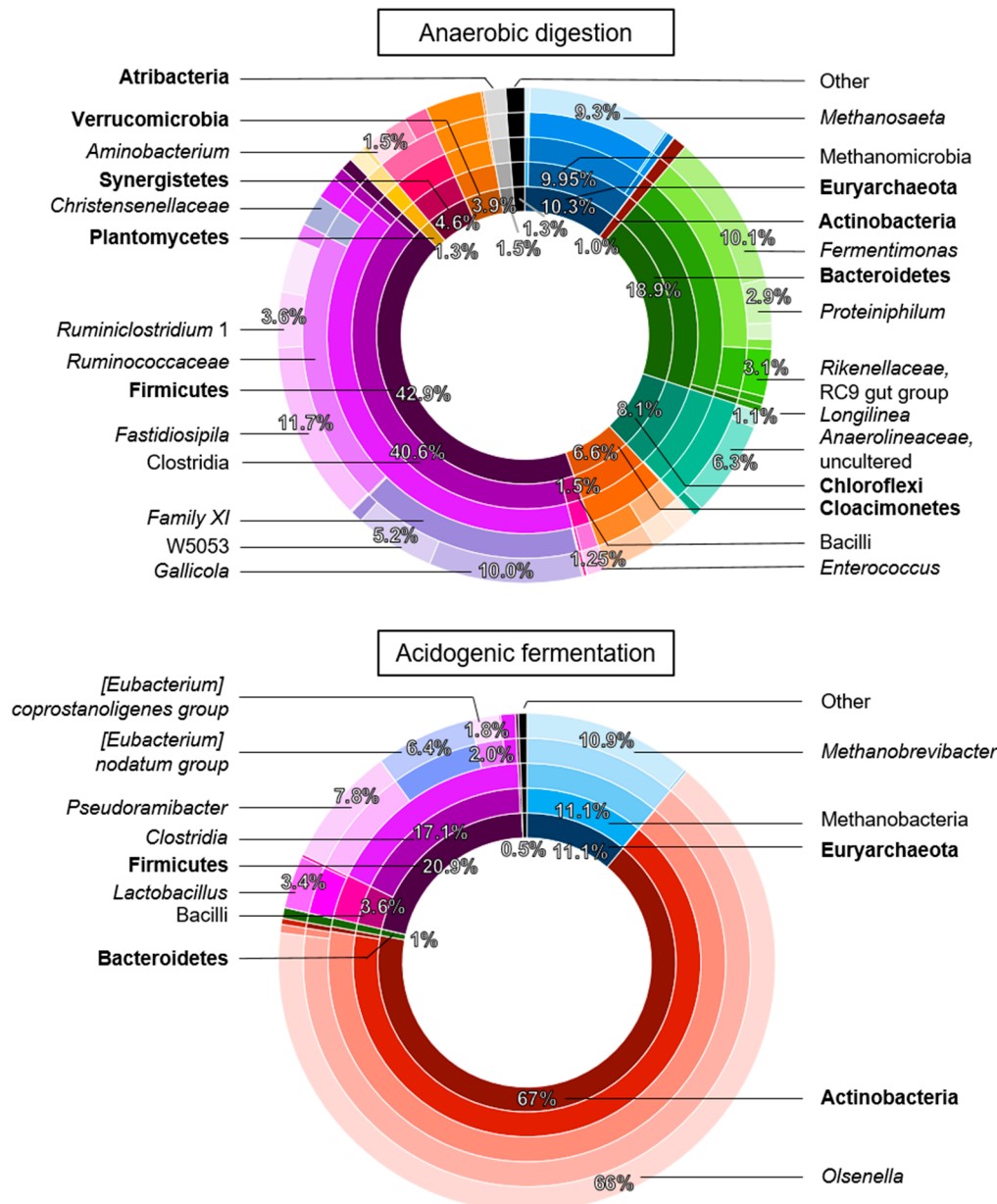

**Figure 4.** Taxonomic composition of the bacterial and archaeal community by the end of Phase 2: the anaerobic digestion reactor on Day 77 (**top**) and, the acidogenic fermentation reactor on Day 84 of operation (**bottom**). The concentric circles represent the taxonomic classification from phylum (bold, centre ring) to genus (outer ring), colours represent classification, and bandwidth and percentages relate to relative abundance.

The community during AF was dominated by Actinobacteria, comprising predominantly the genus *Olsenella* (66 ± 4% over 8 OTUs in AF, <0.1% in AD). The acidotolerant *Olsenella* sp. and the highly diverse genus of *Lactobacillus* (3 ± 2% over 13 OTUs in AF, <0.1% in AD) are linked with hydrolysis and the acidogenic production of, for instance, lactic acid, acetate, $CO_2$, and ethanol from hexoses and pentoses [66–68]. In addition, these genera have been correlated with lactic acid-based chain elongation. *Olsenella* spp. have been found in reactors processing lignocellulosic substrates in co-occurrence with bacteria from the chain elongating genus *Pseudoramibacter*, the third most abundant genus (7.83 ± 0.08%) in our reactor [35,36,69]. *Pseudoramibacter* spp. produce VFA, MCCA and $H_2$ by fermenting carbohydrates, and, as recently suggested, glycerol, and lactic acid [70,71]. *Lactobacillus* spp.

have been detected in FW fermentation alongside *Caproiciproducens* spp., which only had a low relative abundance of 0.53 ± 0.09% in our reactor [37]. Other abundant genera in AF were from the order of Clostridiales. The genus classified as the *Eubacterium nodatum* group (6.4 ± 0.5%) is known for the decomposition of organic matter into VFA and was found before to compete with chain elongation bacteria resulting in excessive C4 production in a xylan-fed fermentation [69]. The genus *E. coprostanoligenes* group (1.8 ± 0.3%) has been found before in FW digesters producing C2, succinic acid and $H_2$ and has a phospholipase activity to reduce cholesterol [60,72].

In terms of the archaeal community, the relative abundance of methanogenic Euryarchaeota was similar in the AD and AF reactor (Figure 4). In the AD reactor, the dominant genus the acetoclastic methanogen *Methanoseata* in AD, which is typical for full-scale AD of organic solid waste [73–75]. In contrast, in AF the hydrogenotrophic *Methanobrevibacter* was the lead methanogenic genus. Hydrogenotrophic methanogens have been found to increase in relative abundance during organic overloading of AD systems, and they are generally more tolerant to environments with high carboxylic acids content [8,76]. This could explain why methane in the AF reactor increased again from Day 56 (Phase 2) onwards, despite the high carboxylic acid concentration.

## 4. Discussion

The functionality of the AD seeding sludge shifted to AF when starting reactor operation at a higher F/M and OLR compared to traditional AD. This is consistent with the response of an anaerobic microbiome to a high organic load, whereby accumulation of VFA inhibits methanogenesis [19,77]. Increased substrate availability by operating at a higher OLR (and indirectly F/M), while maintaining HRT (i.e., increased COD in the FW2) shifted product outcome from methane to VFA, and from short VFA to MCCA. Chain elongation improved, despite a decreased supply of electrons donors in the influent (i.e., FW2). Normally, the elongation of VFA to MCCA via the reversal of the β-oxidation pathway becomes less thermodynamically favourable with less electron donors available [53,78]. However, Arslan, et al. [79] reviewed several mixed culture AF studies and showed that an increase in organic load generally resulted in a more reduced product spectrum of carboxylic acid. During FW fermentation, the electron donors for chain elongation can be produced in situ [37]. The co-occurrence of lactate producing *Olsenella* spp. and lactate consuming chain elongation bacteria with increased availability of substrates confirms that the AF microbiome was able to produce the electron donors required for chain elongation in situ, alongside using the few electron donors in the feedstock.

Increased availability of organics, either by high F/M or OLR, lead to VFA accumulation, which in the case of AD jeopardised the main goal of the process. Doubling the HRT, and thus decreasing OLR, temporarily restored the functionality of the AD reactor. However, in the long-term, working at maximum AD capacity accumulated n-propionic acid, difficult to degrade, which inhibited methanogenesis. Thus, whereas AD requires a more dilute FW feedstock or operation with extended retention times to allow mitigation of OLR stress, high-COD FW streams lend themselves well as feedstock for MCCA production as they allow accumulating electron donors from primary fermentation.

The overall values for C6 are higher than reported for similar un-supplemented STR setups; for instance, 8.5 g $L^{-1}$ C6 were produced with switch-grass stillage feedstock [36]. The concentrations in the current study were closer to those fermenting FW using more sophisticated reactor set-ups, such as leach bed reactors (9.9 g $L^{-1}$ C6) [31], or two-stage ethanol-supplemented up-flow anaerobic reactors (12.6 g $L^{-1}$) [28]. Higher concentrations of 23 ± 1 g $L^{-1}$ C6 have been reported but only when chain elongation was further stimulated by using pre-treated FW, ethanol supplementation and a microbiome previously enriched with synthetic media [80]. However, these FW substrates might differ in composition such as solid or COD content, similar, as FW 1 was different from FW 2. For fermentation of acid whey, it has recently been found that the quality of the feedstock had significant impact on MCCA production [81]. Further research should evaluate the impact of FW composition as the application of this technology would have to deal with the inherent variability of a FW substrate caused by differences in sources, collection, and storage [82,83].

Concentrations of C8 have reached higher concentrations in similar reactor set-ups using alternative feedstocks, e.g., 3.2 g $L^{-1}$ fermenting thin stillage and beer [84] and 3.1 ± 0.9 g $L^{-1}$ for diluted cheese whey powder [85]. These specific feedstocks are high in ethanol from beer and lactic acid from whey, creating a more reductive environment that could stimulate further chain elongation. Thus, better operational control to ensure reductive conditions could enhance the elongation process. For instance, an airtight feeding strategy could improve maintaining a sufficient $p\mathrm{H}_2$. Kinetic studies between feed events indicated that C4–C8 compounds were partially degraded right after a feed event, likely due to loss of $p\mathrm{H}_2$. Future work should also evaluate whether application of a semi-continuous feeding pattern instead of continuous, i.e., subjecting the microbiome to a fluctuating substrate availability, maximizes the benefits of consecutive fermentation stages as seen in batch-like operation. Namely, they stimulate the initial, rapid accumulation of electron donors by primary fermentation and, thus, ensure reductive conditions for consecutive chain elongation.

For anaerobic microbiomes, it is generally regarded that in response to operational changes the hydrolytic and acidogenic bacteria are sufficiently dynamic and able to maintain overall functionality by replacing one another, whereas methanogenic activity will subside after operational disturbance and potentially rebound later on [86]. This typical quality of open fermentation systems where multiple distinct microorganisms are capable of performing similar biochemical function, i.e., functional redundancy, is seen as advantageous since it allows to stabilize reactor functionality following operational perturbation, e.g., substrate fluctuations, or adapt to new environmental conditions [87,88]. Indeed, different genera were responsible for the main functionalities of hydrolysis and acidogenic fermentation in the AD and AF reactors. Thus, these key metabolic functionalities required for dealing with a complex feedstock in AD were maintained in AF. However, the same concept allowed for resilient methanogenic activity by the development of hydrogenotrophic methanogens in the AF reactor. Since a decrease of $p\mathrm{H}_2$ due to hydrogenotrophic methanogenic activity could potentially compromise MCCA yields, the implications for full-scale long-term application should be further evaluated.

The lower pH, i.e., more acid-stress, and shorter retention times in the AF reactor, reduced degradation of the FW solids, as seen by the lower VS removal and reshaped the microbial community of AF into a more specialised and homogenous community than in the AD reactor. This decrease in community richness, diversity and evenness is in line with what other studies reported for organic overloading of AD and for chain elongation studies producing C8 [76,84,89]. Improvements in hydrolysis and MCCA yield, or integration within a broader biorefinery context that includes post chain elongation treatment will be required in practise if equivalent waste reduction and stabilization to AD are to be obtained.

High investments in highly specific infrastructure and/or lack of skills and expertise are some of the main technical barriers hindering adoption of advanced wastewater treatment technologies [90,91]. Here we have shown that MCCA production can be stimulated from FW fermentation without supplementation of methanogenic inhibitors, electron donors or growth medium in a simple, single-stage STR by manipulation of the organic load. Thus, it is similar to the current operation of established AD systems, and in particular it is comparable to acid-phase digesters that also operate at increased organic load and lower pH. In addition, the high-COD FW substrate that required extended retention times or dilution for AD, is more advantageous to apply as substrate for MCCA production hence overcoming some of the difficulties faced in FW AD. However, while the fermentation process itself could allow repurposing current digesters, significant research efforts are still required regarding separation and purification of MCCAs to obtain marketable products.

**Author Contributions:** Conceptualization, M.C.; methodology, V.D.G. and M.C.; investigation, V.D.G. and M.C.; data curation, V.D.G.; writing—original draft preparation, V.D.G.; writing—review and editing, M.C., T.C.A., D.J.L., A.B.L. and V.D.G.; visualization, V.D.G.; supervision, M.C., T.C.A., D.J.L. and A.B.L.; project administration, T.C.A. and A.B.L.; funding acquisition, T.C.A. All authors have read and agreed to the published version of the manuscript.

**Funding:** This research has received funding from the European Union's Horizon 2020 research and innovation programme under the Marie Skłodowska-Curie grant agreement H2020-MSCA-CO-FUND, No 665992. We would like to thank the support and funding from Wessex Water and GENeco (Bristol, UK). This work is supported by EPSRC under grant EP/L016354/1.

**Acknowledgments:** The authors would like to thank Tom Phelps, Ian Law and Wesley Wong from GENeco for enthusiastic support of the research project, and in the reliable and generous supply of samples and information.

**Conflicts of Interest:** The authors declare no conflict of interest.

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
