# Peer review of "Adjusting Organic Load as a Strategy to Direct Single-Stage Food Waste Fermentation from Anaerobic Digestion to Chain Elongation"

_processes, doi:10.3390/pr8111487_

Round 1

Reviewer 1 Report

The article: “Adjusting organic load as a strategy to direct single-stage food waste fermentation from anaerobic digestion to chain elongation” is interesting and may be of interest to Processes readers. In order to improve the text quality, recommends minor revision.

Comments:

  • Please complete the introduction with literature items from the last years 2019-2020.
  • The novelty of the work is not clearly explained in relation to the work of other researchers.
  • Please clearly emphasize the purpose of the work
  • The reference and description to Table 1 should be more specific
  • The quality Fig 1-3 should be improved or enlarged because they are difficult to read.
  • In the Discussion section,
  • -expand the following sentence: “Indeed, the main functionalities of hydrolysis and acidogenic 396 fermentation were present in both the AD and AF reactors, but by delivered by different genera” Determine whether the phenomenon has a positive or negative effect.
  • There is no concrete summary of the research carried out in the context of its practical application in Production renewable feedstock

Author Response

I am very pleased to submit the revised manuscript of our research article “Adjusting organic load as a strategy to direct single-stage food waste fermentation from anaerobic digestion to chain elongation”.

We would like to thank the reviewers for their time and consideration. Their helpful comments and positive feedback were much appreciated. We have carefully considered the reviewers’ comments and revised the manuscript accordingly. A response to comments from Reviewer 1 can be found below.

We thank you for your consideration.

General comment: The article: “Adjusting organic load as a strategy to direct single-stage food waste fermentation from anaerobic digestion to chain elongation” is interesting and may be of interest to Processes readers. In order to improve the text quality, recommends minor revision.

Response: We thank the reviewer for these positive comments. The feedback on how to elevate the quality of the manuscript is much appreciated.

Here we respond to each of the comments from the reviewer in turn, and we would like to point out that when we indicate line numbers, this is on the assumption that the track-changes annotations in the manuscript have all been accepted, ie with respect to a clean version of our second version of the manuscript.

Comment 1: Please complete the introduction with literature items from the last years 2019-2020.

Response 1: We have added the following citations to provide examples of more recent reviews on MCCA production by mixed culture fermentation (L50), and to share more recent developments in acidogenic fermentation for lactic acid, hydrogen and VFAs production (L52):

  • Han, W.; He, P.; Shao, L.; Lü, F. Road to full bioconversion of biowaste to biochemicals centering on chain elongation: A mini review. Journal of Environmental Sciences 2019, 86, 50-64, doi:10.1016/j.jes.2019.05.018.
  • De Groof, V.; Coma, M.; Arnot, T.; Leak, D.J.; Lanham, A.B. Medium Chain Carboxylic Acids from Complex Organic Feedstocks by Mixed Culture Fermentation. Molecules 2019, 24, doi:10.3390/molecules24030398.
  • Greses, S.; Tomás-Pejó, E.; González-Fernández, C. Short-chain fatty acids and hydrogen production in one single anaerobic fermentation stage using carbohydrate-rich food waste. Journal of Cleaner Production 2020, 10.1016/j.jclepro.2020.124727.
  • Xu, X.; Zhang, W.; Gu, X.; Guo, Z.; Song, J.; Zhu, D.; Liu, Y.; Liu, Y.; Xue, G.; Li, X., et al. Stabilizing lactate production through repeated batch fermentation of food waste and waste activated sludge. Bioresour Technol 2020, 300, 122709, doi:10.1016/j.biortech.2019.122709.
  • Im, S.; Lee, M.-K.; Yun, Y.-M.; Cho, S.-K.; Kim, D.-H. Effect of storage time and temperature on hydrogen fermentation of food waste. International Journal of Hydrogen Energy 2020, 45, 3769-3775, doi:10.1016/j.ijhydene.2019.06.215.

In addition, thanks to the reviewer’s suggestion, we discovered some recent work on improvement of MCCA production by different pretreatment methods that was new to us, which has been added (L80-81):

  • Ma, H.; Lin, Y.; Jin, Y.; Gao, M.; Li, H.; Wang, Q.; Ge, S.; Cai, L.; Huang, Z.; Van Le, Q., et al. Effect of ultrasonic pretreatment on chain elongation of saccharified residue from food waste by anaerobic fermentation. Environmental Pollution 2020, 10.1016/j.envpol.2020.115936, doi:10.1016/j.envpol.2020.115936.

Comment 2: The novelty of the work is not clearly explained in relation to the work of other researchers.

Response 2:

We have included new text in the Abstract (L34-36) and Introduction (L96-103) to clarify the novelty of the work, which relates mainly to the opportunity to use manipulation of feeding strategy to allow repurposing of existing AD reactor assets from biogas production to formation of higher value MCCA products.

Comment 3: Please clearly emphasize the purpose of the work

Response 3: The final paragraph of the introduction has been re-written to improve clarity on the aim of the research (L96-103). Also please see Response 2 above.

Comment 4: The reference and description to Table 1 should be more specific

Response 4: The description (L 105-114), caption (L 115-116) and subtext of Table 1 have been extended to ensure all information is provided and clarified. The text now includes a more detailed description of the substrate’s origin and how it was processed.

Comment 5: The quality Fig 1-3 should be improved or enlarged because they are difficult to read.

Response 5: The figures have been replaced by higher quality, larger figures with increased font size. We hope these modifications have improved legibility.

Comment 6: In the Discussion section, expand the following sentence: “Indeed, the main functionalities of hydrolysis and acidogenic fermentation were present in both the AD and AF reactors, but by delivered by different genera”. Determine whether the phenomenon has a positive or negative effect.

Response 6: This is an interesting question as this phenomenon has both a negative and positive effect to it. It is positive in terms of maintaining hydrolytic and primary fermentative functions, yet negative when it comes to the resilience of methanogenic activity in AF. We have rewritten the text to include the discussion on this (L411-424).

Comment 7:  There is no concrete summary of the research carried out in the context of its practical application in Production of renewable feedstock.

Response 7: We agree that a better discussion including practical application would improve the manuscript. Thus, we have re-written and extended the final paragraphs of the discussion (L428-442).

Reviewer 2 Report

The manuscript is well written and carried experiments out systematically; however, adjusting OLR as a strategy to the AD/AF for MCCA production from food waste is quite general study. Few suggestions are given as follows:

  1. It would be good if this study just focused on evaluating varied operational parameters of AF for MCCA production so the AD tests would not be needed 
  2. In Discussion, when the study was compared to other studies, besides the performance, the authors should give some explanation on the different operational parameters that could also show different or similar performance, for example:
    • P12, L374: Is it positive or negative that the performance of both un-supplemented STRs was similar but using totally different feedstocks (FW and lignocellulosic biomass)?
    • P12, L376-L378: Even though the efficiency was similar to others, there is difference in the composition of substrate (e.g. %TS, COD conc, lignin content) and HRT

Author Response

I am very pleased to submit the revised manuscript of our research article “Adjusting organic load as a strategy to direct single-stage food waste fermentation from anaerobic digestion to chain elongation”.

We would like to thank the reviewers for their time and consideration. Their helpful comments and positive feedback were much appreciated. We have carefully considered the reviewers’ comments and revised the manuscript accordingly. A response to comments from Reviewer 2 can be found below.

We thank you for your consideration.

General Comments:
The manuscript is well written and carried experiments out systematically; however, adjusting OLR as a strategy to the AD/AF for MCCA production from food waste is quite general study.

Response: Please see our response to a similar comment from Reviewer 1, specifically our responses to Comments 2 and 3 from Reviewer 1.

Here we respond to each of the comments from the reviewer in turn, and we would like to point out that when we indicate line numbers, this is on the assumption that the track-changes annotations in the manuscript have all been accepted, ie with respect to a clean version of our second version of the manuscript.

Comment 1: It would be good if this study just focused on evaluating varied operational parameters of AF for MCCA production so the AD tests would not be needed

Response 1: We agree a systematic assessment of operational parameters for MCCA production from food waste fermentation would be valuable. Here, we wanted to emphasize how a microbial culture performing AD can be steered towards AF and MCCA production. We wanted to keep the AD reactor in parallel using the same feedstock and inoculum as a positive control. This way we would have an AD microbiome and a functionality profile in comparison to the AF reactor that had been fed with the same feedstock over the same amount of time and in similar conditions (apart from the different operational parameters that were the topic of study). We have rewritten L94-99 in the Introduction to clarify our aim.

Comment 2: In Discussion, when the study was compared to other studies, besides the performance, the authors should give some explanation on the different operational parameters that could also show different or similar performance, for example:

  • P12, L374: Is it positive or negative that the performance of both un-supplemented STRs was similar but using totally different feedstocks (FW and lignocellulosic biomass)?
  • P12, L376-L378: Even though the efficiency was similar to others, there is difference in the composition of substrate (e.g. %TS, COD conc, lignin content) and HRT

Response 2:

We thank the reviewer for this suggestion as indeed, the composition of the substrate is a crucial parameter to include in the discussion. We have, thus, re-written and extend the discussion (L 388-396)